# Methanogenesis and Salt Tolerance Genes of a Novel Halophilic Methanosarcinaceae Metagenome-Assembled Genome from a Former Solar Saltern

**DOI:** 10.3390/genes12101609

**Published:** 2021-10-13

**Authors:** Clifton P. Bueno de Mesquita, Jinglie Zhou, Susanna M. Theroux, Susannah G. Tringe

**Affiliations:** 1Department of Energy, Joint Genome Institute, Lawrence Berkeley National Laboratory, Berkeley, CA 94720, USA; cliff.buenodemesquita@lbl.gov (C.P.B.d.M.); jingliezhou@gmail.com (J.Z.); 2Southern California Coastal Water Research Project, Costa Mesa, CA 92626, USA; susannat@sccwrp.org; 3Environmental Genomics and Systems Biology Division, Lawrence Berkeley National Laboratory, Berkeley, CA 94720, USA

**Keywords:** methanogenesis, salt tolerance, anaerobic, archaea, phylogenomics

## Abstract

Anaerobic archaeal methanogens are key players in the global carbon cycle due to their role in the final stages of organic matter decomposition in anaerobic environments such as wetland sediments. Here we present the first draft metagenome-assembled genome (MAG) sequence of an unclassified Methanosarcinaceae methanogen phylogenetically placed adjacent to the *Methanolobus* and *Methanomethylovorans* genera that appears to be a distinct genus and species. The genome is derived from sediments of a hypersaline (97–148 ppt chloride) unrestored industrial saltern that has been observed to be a significant methane source. The source sediment is more saline than previous sources of *Methanolobus* and *Methanomethylovorans*. We propose a new genus name, *Methanosalis*, to house this genome, which we designate with the strain name SBSPR1A. The MAG was binned with CONCOCT and then improved via scaffold extension and reassembly. The genome contains pathways for methylotrophic methanogenesis from trimethylamine and dimethylamine, as well as genes for the synthesis and transport of compatible solutes. Some genes involved in acetoclastic and hydrogenotrophic methanogenesis are present, but those pathways appear incomplete in the genome. The MAG was more abundant in two former industrial salterns than in a nearby reference wetland and a restored wetland, both of which have much lower salinity levels, as well as significantly lower methane emissions than the salterns.

## 1. Introduction

Methanogens play a key role in the global carbon cycle by performing the final steps of organic matter decomposition into methane (CH_4_) and carbon dioxide (CO_2_) in anaerobic environments. Natural methanogenesis contributes ~217 Tg CH_4_ yr^−1^ to the atmosphere, with wetlands as the single largest natural source of methane at an estimated ~149 Tg CH_4_ yr^−1^ of emissions [1]. Three major pathways of anaerobic archaeal methanogenesis have been described in the literature—acetoclastic (acetate splitting, with acetate acting as an electron donor and acceptor), hydrogenotrophic (using hydrogen, formate, carbon monoxide or alcohols as electron donors and carbon dioxide as an electron acceptor), and methylotrophic (involving demethylation of methylamine, dimethylamine, trimethylamine, dimethyl sulfide, methanethiol, tetramethylammonium, methanol, or glycine betaine) methanogenesis [2]. The relative contribution of these different pathways depends on several variables including organic matter quality and composition, salinity, temperature, and pH [2].

While all three of these pathways are performed by members of the Euryarchaeota phylum, there are some distinctions at the order, family, and genus levels. Methanomicrobiales, Methanobacteriales, Methanocellales, and Methanosarcinales have been described as hydrogenotrophic orders, while the Methanosarcinaceae and Methanotrichaceae families within the Methanosarcinales order contain acetoclastic taxa [3]. Methanosarcinaceae is recognized as the most metabolically diverse family, containing genera that can perform all three pathways [4]. Members of the genera *Methanolobus* and *Methanomethylovorans* (Methanosarcinaceae) are methylotrophic methanogens, as many experiments on pure cultures have shown growth and methane production with methylotrophic substrates but not acetate or hydrogen/carbon dioxide [5,6,7,8,9,10,11,12,13,14,15].

Methanogenesis in saline to hypersaline environments presents an intriguing situation both in terms of extremophilic adaptations for survival and metabolic adaptations to avoid competition. Two families of methanogens, Methanosarcinaceae and Methanocalculaceae, contain halophilic members [16]. Several studies conducted in hypersaline environments, including solar salterns and hypersaline lakes, have identified key halophilic methanogenic genera inhabiting these environments, including *Methanohalobium*, *Methanohalophilus*, *Methanosalum*, *Methanocalculus*, and *Methanosarcina* [16,17,18,19]. These genera not only tolerate extremely high salinity (up to 30%) but grow optimally in hypersaline conditions (15%). By contrast, the genera *Methanolobus* and *Methanomethylovorans* have not been observed in the most hypersaline environments. *Methanomethylovorans* have been isolated from freshwater sediments [10,11,20], while cultivated members of *Methanolobus* appear to be at least halotolerant (growing in up to 10% salinity) [21] but perhaps not true halophiles based on the salt levels where they achieve optimum growth [22]. The methylotrophic pathway likely predominates in hypersaline environments for two reasons: first, sulfate reducers do not compete for methylated compounds as they do for hydrogen or acetate [23,24]; and second, methyl-group containing compounds are available in these environments due to the breakdown of compatible solutes such as glycine betaine and dimethylsulfoniopropionate that are produced by both prokaryotes and algae [25].

Several options exist for archaea in hypersaline environments to achieve the necessary osmoregulation for their survival. These have been broken down into two categories, the “salt in’’ strategy typically used by Haloarchaea, which involves increasing cytoplasm salt concentrations, usually with potassium chloride (KCl), and the “low salt in” strategy (also sometimes referred to as “salt out” strategy), typically used by bacteria but also some archaea, which involves production of compatible solutes to maintain osmotic balance in the cytoplasm without increased salt [26,27]. In particular, other halophilic archaea (i.e., Halobacteriales) have been shown to produce trehalose and betaine, and the genes required for their synthesis and transport have been described [28,29]. Previous studies have also described several other compounds such as glutamate, sucrose, ectoine, and glycosylglycerol as compatible solutes for halotolerance, as well as the major alkali metal-cation transporters that are involved in the “salt in” strategy [27]. However, the presence of genes involved in synthesis and transport of these salts and solutes has not been assessed in the available *Methanolobus* and *Methanomethylovorans* genomes.

In this paper we describe a new draft genome for *Methanosalis* sp. SBSPR1A, gen. nov., sp. nov., with a particular emphasis on methane cycling genes and halotolerance genes. We also compare the genome to closely related *Methanolobus* and *Methanomethylovorans* taxa phylogenetically, as well as in terms of shared and unique orthologous gene groups.

## 2. Materials and Methods

### 2.1. Metagenome Information

The genome was assembled from metagenomic sample R1_A_D2 (IMG ID 3300026157, NCBI BioSample accession SAMN06266236) as part of the study published by Zhou et al. [30] on South San Francisco Bay salt ponds, who initially classified the MAG as *Methanolobus* with Bin Annotation Tool (BAT) [31]. The genome name SBSPR1A is based on the sample from which it is derived (South Bay Salt Pond R1 replicate A). This sample is the 5–15 cm deep section of a sediment core taken from a 57.2 ha former industrial salt pond (37.5° N, −122.13° W) that is no longer used for salt production but has not been restored. Industrial salt production occurred in the pond from the 1850s to 2003. The sample was characterized by high salinity (97.2 total ppt Cl^−^), a temperature of 24.7 °C, low dissolved oxygen content (0.96 mg L^−1^) and a pH of 7.75 [30]. With salinity levels over twice that of seawater (~35 ppt), the pond can be considered hypersaline [25]. Elevated methane emissions were measured from this core in the field (737 µmol CH_4_ m^−2^ d^−1^) and in a lab incubation using sediments collected from the same pond [30]. Approximately 10 Gb of shotgun sequence data were generated from this sample on the HiSeq 2500 (Illumina, San Diego, CA, USA) platform in a 2 × 150 paired-end run mode. Reads were filtered and trimmed with BBTools (https://sourceforge.net/projects/bbmap/, accessed 16 April 2018) and assembled with SPAdes [32].

### 2.2. MAG Assembly

Metagenome assembled genomes (MAGs) were binned using MetaBat [33], MaxBin [34], and CONCOCT [35], the best of which were selected using DAS Tool [36]. For this draft genome, we began with a MAG binned by CONCOCT, with a completeness of 98.37%, contamination of 3.27%, and strain heterogeneity of 0, as calculated by CheckM [37], meeting community standards for a “high-quality draft” [38].

Abundance estimates were calculated by mapping the raw metagenomic reads to the MAG, taking the average read depth, and then transforming to counts per million of assembled reads to take into account differences in library sizes across the 24 metagenome samples [30]. The effect of site (4 separate wetlands) on MAG abundance was calculated with a Kruskal-Wallis test (R package stats) followed by Nemenyi posthoc (R package PMCMR [39]), as assumptions for ANOVA and Tukey HSD were not met (Levene Test, *p* < 0.05, Shapiro-Wilk test, *p* < 0.05).

Scaffolds were extended manually using Geneious [40] to remap paired-end reads to each scaffold and extend using reads partially overlapping the ends of the scaffolds [41]. For scaffolds that could not be extended using reads from sample R1_A_D2, reads from sample R1_A_D1, the 0–5 cm section of the same core, were used. After each round of scaffold extension, the new scaffold set was assembled de novo with the Geneious assembler to merge any newly overlapping scaffolds, and then reads were remapped to the reassembled scaffolds to resolve any ambiguous bases created during the assembly. Following two rounds of scaffold extension, we attempted to close two gaps (one group of 71 Ns and one group of 10 Ns) with both Sealer [42] and GapCloser from SOAPdenovo2 [43], but neither program was able to close either gap. The genome was then uploaded to the Joint Genome Institute’s Integrated Microbial Genomes and Microbiomes database (IMG/M) [44] for gene annotation, and is publicly available with IMG ID 2929001634 and on GenBank under BioProject ID PRJNA365332.

### 2.3. Comparative Genomics

For comparative genomics, we downloaded nine *Methanolobus* reference genomes available from the RefSeq database, representing the species *Methanolobus psychrophilus* R15 (NC_018876) [15], *Methanolobus psychrotolerans* YSF-03 (NZ_MBKP01000078) [21], *Methanolobus tindarius* DSM 2278 (NZ_AZAJ01000001) [45], *Methanolobus vulcani* B1d (NZ_VIAQ01000006) [46], *Methanolobus vulcani* PL 12/M (NZ_FNCA01000018) [47], *Methanolobus profundi* Mob M (NZ_FOUJ01000018) [8,48], *Methanolobus* sp. SY-01 (NZ_PGGK01000010) [49], *Methanolobus zinderi* DSM 21339 (NZ_CP058215) [6], and *Methanolobus bombayensis* DSM 7082 (NZ_JAGGKD010000001) [9,50]. We also downloaded the one *Methanomethylovorans* genome on RefSeq, *Methanomethylovorans hollandica* DSM 15978 (NC_019977) [10,45]. We first used the “Insert Set of Genomes Into SpeciesTree” tool (version 2.2.0) in KBase [51] to place these 11 genomes into a phylogenetic tree with 50 other genomes from RefSeq [52] using 49 single copy clusters of orthologous groups (COGs). Next, a more detailed phylogenetic tree was constructed by first identifying and aligning 122 universal single copy archaeal marker genes with GTDB-Tk [53] and then building a consensus tree from the concatenated alignment with RAxML version 8.12 [54] with the PROTGAMMALG model of amino acid substitution, 1000 bootstraps, and *Nitrosopumilus adriaticus* (RefSeq GCF_000956175.1) as an outgroup. We used ProtTest 3 [55] to select the best model of amino acid substitution (LG). Furthermore, we constructed a phylogenetic tree of full length or near full length 16S rRNA genes, which were acquired from IMG where available or extracted with ContEst16S [56], by aligning with MUSCLE [57] and then using the IQ-Tree webserver to build the tree [58]. Some additional taxa with available 16S rRNA genes on NCBI were added to this analysis—*Methanolobus oregonensis* (U20152.1), *Methanolobus taylorii* (U20154.1), *Methanolobus chelungpuianus* (EU293796.1), *Methanomethylovorans uponensis* EK1 (KC876048.1), *Methanomethylovorans thermophila* (AY672821.1), *Methanomethylovorans victoriae* TM (AJ276437.1), and *Methanomethylovorans* sp. Z1 (EF174501.1). Pairwise percentage similarity of these 16S rRNA gene sequences was calculated with NCBI BLAST. Average nucleotide identity (ANI) was calculated with FastANI implemented in KBase. The KBase narrative including the ANI analysis and phylogenetic analysis for this project is publicly available under narrative ID 90991.

To ensure consistent gene annotations across genomes, Prodigal [59] was used to predict protein-coding genes from the nucleotide sequences. Mean isoelectric points of the protein-coding sequences were calculated with EMBOSS [60] using the “iep” tool on the Galaxy website platform (https://usegalaxy.org/, accessed 28 July 2021) [61] with a pH step of 0.5. Analysis of orthologous gene groups was conducted with proteinortho [62] and KEGG orthology (KO) profiles [63] and intersections among the genomes were graphed with the ComplexUpset R package [64]. KO profiles were downloaded from IMG where available; for genomes not yet on IMG, KO profiles were created with BLASTKoala [65]. Selected genes from proteinortho were classified with Pfam [66].

The KO tables were queried for KOs involved in methanogenesis and salt tolerance. A master map of methanogenesis processes containing the genes, enzymes, and reactions of all three pathways, as well as synthesis of precursor molecules such as coenzyme B and coenzyme M and the regeneration of these molecules from heterodisulfide after methane production, was built by synthesizing information in the ModelSeed [67], KEGG [68], and BioCyc [69] databases and mapped with Escher in ModelSeed (Appendix A). A list of KOs was created by utilizing existing KO associations with reactions in these databases, although not all of the reactions were annotated with KOs as enzymes have not yet been identified for some reactions. A list of KOs involved in salt tolerance was developed based on salt tolerance related genes and processes described in the literature [22,26,27,28,29,70]. These KOs include proteins involved in compatible solute biosynthesis and transport (e.g., betaine, trehalose, choline, proline), as well as other cation transporters. Presence or absence of these methanogenesis and salt tolerance related KOs were plotted as heatmaps using the pheatmap R package [71]. All R analyses were performed with version 4.0.2 [72].

## 3. Results

The draft genome of *Methanosalis* sp. SBSPR1A consists of 2,476,202 bp in 48 scaffolds, with 2656 protein-coding genes and a GC content of 39.6%. GTDB-Tk classified the genome to the genus level as *Methanolobus*, in agreement with the original MAG classification by Bin Annotation Tool. However, using 49 single copy COGs to place the genomes into a reference tree, *Methanosalis* sp. SBSPR1A is placed between the *Methanomethylovorans* and *Methanolobus* genera (Appendix A). Similarly, using the set of 122 universal archaeal single copy protein-coding genes in GTDB-Tk, the genome is again placed on its own branch between these two genera (Figure 1). This result is corroborated when just considering the full length 16S rRNA gene phylogenetic analysis (Appendix A). The 16S rRNA gene percent similarity between *Methanosalis* sp. SBSPR1A and the *Methanomethylovorans* taxa was between 90.40–91.78% and compared to the *Methanolobus* taxa was between 91.68–92.69% (Appendix A). These values are below 94.5% and are suggestive of a new genus and species [73]. According to ANI, the *Methanosalis* sp. SBSPR1A genome is most closely related to *Ml. bombayensis*, but differences in pairwise ANI between this genome and those of the other taxa were small (Appendix A). ANI values between *Ms*. sp. SBSPR1A and the other genomes ranged between 76.97 and 77.97, again suggesting that it is a unique genus and species (Appendix A). MASH ANI clustering generally agrees with the marker gene clustering with the exception that *Methanosalis* sp. SBSPR1A groups more closely with *Mmv. hollandica* and the *zinderi*-SP01-*psychrophilus Methanolobus* clade in MASH ANI clustering (Appendix A).

There were 3772 orthologous gene groups identified across all the genomes computed by proteinortho; using KEGG Orthology annotations, there were a total of 1439 KOs across the eleven genomes. This is consistent with about 45% of the protein-coding genes in these genomes having a KO annotation. Among the eleven genomes examined here, according to proteinortho there were 1347 orthologous genes shared among all eleven taxa and 61 genes present in the other ten genomes but not the *Methanosalis* sp. SBSPR1A genome (Figure 2a). The genes missing from our focal genome include a wide variety of functions (Appendix A). A small number of genes were found only in *Ms*. sp. SBSPR1A and one other genome; among these, the largest number (13) was shared with *Ml. profundi*. For KEGG orthology gene groups, there were 15 KOs present in the other 11 genomes but not that of *Ms*. sp. SBSPR1A (Figure 2b). Of these, a few notable genes include *comDE* for biosynthesis of coenzyme M, an important precursor to methane, and *mtaBC* for methanogenesis from methanol (Appendix A). There were 18 KOs present unique to *Ms*. sp. SBSPR1A. Of these, functions included ABC transporters, *sdmt* and *gsmt* betaine biosynthesis genes, ribosomal protein *rplT*, and the purine metabolism *ADK* adenosine kinase gene (Appendix A).

In terms of methane cycling, all of the genomes contained the three key *mcrABG* genes necessary for methyl-CoM reduction to methane, the terminal step in all of the archaeal methanogenesis pathways (Figure 3 and Appendix A). As for the pathways for methyl-CoM formation, at least some genes involved in each of the acetate, hydrogen/CO_2_, and methylotrophic pathways were present in all of the genomes. However, the acetoclastic and hydrogenotrophic routes appear to be incomplete in the MAG and the *Methanolobus* and *Methanomethylovorans* genomes, with all 11 genomes missing *ackA*, *pta*, and *eutD* in the acetate pathway and missing *fwdH* and *hmd* in the hydrogenotrophic pathway. *Methanosalis* sp. SBSPR1A contained all of the genes for demethylation pathways of trimethylamine and dimethylamine, but only some of the genes for methylamine and methanol demethylation, while several of the other taxa contained all of the genes for all four of the methylotrophic pathways with KO annotations (Figure 3). Note that methyl-CoM can also be produced from methanethiol, dimethyl sulfide, tetramethylammonium, and glycine betaine [69], but these pathways did not have KOs assigned to the biochemical reactions as of the time of writing. During methyl-CoM reduction to methane, heterodisulfide is produced as a byproduct, which can then be cycled back into CoM and CoB via several different pathways (Appendix A). Genes involved in these pathways are patchily distributed among the 11 genomes, with the pathway using cofactor F_420_ being the only consistently complete pathway (*hdrA2*, *hdrB2*, *hdrC2* genes).

Additionally, upstream of the pathways discussed above, two compounds must be synthesized or imported—coenzyme B (CoB) and coenzyme M (CoM) [74], the latter of which can be synthesized from phosphoenol pyruvate (CoM synthesis I) or O-phosphate-L-serine (CoM synthesis II) [69]. Each of these processes involves multiple chemical reactions, only some of which are currently annotated with KOs. Interestingly, even with the limited number of described KOs involved in these pathways, all 11 of the genomes analysed here contain less than half of those described KOs (Appendix A). The KOs present in the genomes are consistent, with the exception of *comDE* for CoM synthesis which is missing from *Ms*. sp. SBSPR1A.

Another interesting phenotype of this taxon is its halotolerance, as it was sequenced from a sample with 97 ppt salinity and was also abundant in samples containing up to 232 ppt salinity. Similar to other halotolerant archaea [75], *Ms*. sp. SBSPR1A had an acidic proteome, with the protein-coding sequences having a mean isoelectric point (pI) of 5.90 ± 0.04 SE. Similarly, the other *Methanolobus* and *Methanomethylovorans* genomes analyzed here had mean isoelectric points ranging from 5.73 to 6.50 (Figure 1). Isoelectric point profiles for all 11 genomes were asymmetrically bimodal, with a large peak around pH 4–5 and a second but much smaller peak around pH 9–10 (Appendix A). The profiles of the *Methanolobus*, *Methanomethylovorans* and *Methanosalis* genomes were all similar despite *Ms*. sp. SBSPR1A being found in a more hypersaline environment than the other ten species; *Ms*. sp. SBSPR1A also did not have the most acidic proteome. *Ms*. sp. SBSPR1A was present in all 24 sediment samples we characterized from the South San Francisco Bay but significantly more abundant in the 12 unrestored saltern samples than in the reference wetland and restored saltern which had much lower salinities (Nemenyi posthoc *p* < 0.05, Figure 4). While other *Methanolobus* taxa have also been found to be moderately halotolerant, tolerating salinities of up to 117 ppt NaCl, their optimum growth was reported to range from 0 to 35 ppt NaCl, which is the approximate salinity of the ocean [21]. *Methanomethylovorans hollandica* and other *Methanomethylovorans* species grow optimally between 0–6 ppt NaCl and only tolerate up to 17.5 ppt NaCl [10,11,12]. Thus, the *Methanosalis* genome sequenced here comes from a sample with salinity well over both the optimum conditions and upper tolerances reported in *Methanolobus* and *Methanomethylovorans*.

*Ms*. sp. SBSPR1A was the only genome with the *sdmt* and *gsmt* genes for betaine biosynthesis (Figure 5). Nine of the genomes contained at least one gene for betaine transport, the exceptions being *Ml. psychrophilus* and *Mmv. hollandica*. All 11 genomes contained *TrkA* potassium uptake uniporter genes, *kpdB* ATP-driven potassium transport system, *lysC* and *asd* for ectoine biosynthesis, and *proABC* for proline biosynthesis. Some genomes additionally contained genes for sucrose (*SPP*) and glutamate (*gdhA*) biosynthesis. Thus, there appear to be some differences among the taxa in compatible solute strategies.

## 4. Discussion

*Methanolobus* and *Methanomethylovorans* taxa have previously been isolated from diverse environmental samples encompassing a range of conditions, but the organism whose genome is presented here originates from a former saltern sediment sample with higher salinity than any samples in which those genera have previously been found [21]. Interestingly, our saline 24.7 °C sample harbored a Methanosarcinaceae species closely related to *Methanolobus* species isolated from Tibet [15] and Siberia [21], both of which experience below freezing temperatures, demonstrating a broad temperature range for this subgroup of Methanosarcinaceae. At another set of extreme conditions, *Ml. zinderi* was isolated from deep sea vents [6] and *Ml. profundi* was isolated from deep subsurface sediments in a natural gas field [8]. Despite these different habitats, Methanosalis sp. SBSPR1A and all of the *Methanolobus* species, as well as *Methanomethylovorans hollandica*, possess generally the same architecture for methylotrophic methanogenesis and halotolerance. *Ms*. sp. SBSPR1A has the highest diversity of halotolerance genes among the genomes examined but is missing some genes for methanol and methylamine demethylation that are present in most of the other genomes.

*Methanolobus* and *Methanomethylovorans* taxa have previously been described as methylotrophic methanogens. Methylotrophic methanogenesis involves the demethylation of methylated compounds to produce methyl-CoM, followed by reduction of methyl-CoM to methane (Appendix A). Six of the 11 genomes contained genes for demethylation of trimethylamine, dimethylamine, methylamine, and methanol, which are the four molecules with demethylation reactions currently annotated with KEGG K numbers. These genomic data agree with experimental results for these taxa [5,6,7,8,14,15,76]. Interestingly, the other five genomes, including the new genome described in this paper, were missing some of these methanogenesis KOs. Most notably, the genomic data alone suggest that *Methanosalis* sp. SBSPR1A might not perform methanogenesis from methanol and methylamine, only from trimethylamine and dimethylamine. This agrees with experimental data from a sediment core from this pond (but note that this was not a pure culture of *Methanosalis* sp. SBSPR1A) in which methane emissions increased more, and more rapidly, when trimethylamine was added compared to methanol or acetate [30]. Methanol is a product of pectin degradation [77], which proceeds slowly in lake sediments [77]; perhaps for this reason methanol was only a marginal methanogenic precursor in a previous lake sediment study [78] and may not be as important as trimethylamine in the salterns or wetland studied here. It is also possible that these genes are missing from the genome because it is not 100% complete, and this should be confirmed experimentally with pure cultures. Similarly, *Methanolobus* sp. SY-01, *Ml. psychrotolerans*, *Ml. vulcani* B1d, and *Mmv. hollandica* might not be able to perform methanogenesis from all four methyl-containing compounds examined here. *Mmv. hollandica* was previously shown to use dimethyl sulfide and methanethiol [10].

Methylotrophic methanogenesis is an important carbon cycling pathway in saline environments [2,25]. While on a global scale it contributes less methane than acetoclastic or hydrogenotrophic methanogenesis, it can be an important local carbon cycling pathway [79,80,81]. Saline environments with high sulfate concentrations suppress acetoclastic and hydrogenotrophic methanogenesis due to the activity of sulfate reducers, which outcompete methanogens for substrates such as acetate and hydrogen; methylotrophic methanogens avoid this competition [2,23,82,83]. Furthermore, the compatible solutes used to tolerate the saline environments can also be degraded to methyl-containing compounds, particularly trimethylamine and dimethyl sulfide, thus providing a source of substrates for methylotrophic methanogenesis [2]. For example, betaine can be fermented to produce trimethylamine [84], while the osmolyte dimethylsulfoniopropionate can be converted to methanethiol and dimethyl sulfide [85]. Thus, both high salinity and high sulfate environments have been shown to differentially select for methylotrophs over acetoclastic and hydrogenotrophic taxa [86,87]. More specifically, in this particular saltern, due to the abundance of betaine, trimethylamine may be more abundant than methanol, thus supporting an organism such as *Ms*. sp. SBSPR1A that likely performs methanogenesis from trimethylamine but not methanol.

The *Ms*. sp. SBSPR1A genome contains genes involved in compatible solute biosynthesis and transport, which may explain its ability to adapt to hypersaline former solar salterns. The data presented here do not clearly determine which of the two “salt in’’ or “low salt in” strategies *Ms*. sp. SBSPR1A uses. In fact, the presence of KOs involved in both compatible solute synthesis and transport, as well as potassium and sodium transporters suggest that both strategies could be at play, as has been discussed for other organisms [75,88]. Increasing salinity levels have been associated with increasing proteome acidity [89,90]. *Ms*. sp. SBSPR1A has an acidic proteome as has been reported for other halotolerant archaea and bacteria including *Haloferax*, *Halorubrum*, *Halobacterium*, *Salinibacter*, *Halomonas*, and *Alteromonas* [70,75,90]. A mean isoelectric point value of 5.90 (and range of 5.73–5.97 for the other *Methanolobus* genomes excluding *Ml. psychrophilus*, *Ml*. sp. SY-01, and *Ml. psychrotolerans*) is similar to values reported for *Halobacterium* NRC-1 (5.03) and *Salinibacter ruber* (5.92), which are thought to use the “salt in” strategy [75]. However, it has also been shown that organisms such as *Halorhodospira halophila*, *Halomonas elongata*, or *Chromohalobacter salexigens* that produce compatible solutes (i.e., “low salt in” strategy) can also have an acidic proteome and high intracellular KCl concentrations (typical of a “salt in” strategy) [88,91]. Regardless of the strategy used, the mean p*I* values and p*I* profile of *Ms*. sp. SBSPR1A is more similar to moderately and extremely halophilic organisms than to other freshwater, ruminant, or thermophilic methanogens (e.g., mean isoelectric points of 6.32–6.83 [92], symmetrical bimodal profile of *Methanococcus jannaschii* [93]), perhaps highlighting the adaptation of *Ms*. sp. SBSPR1A to the conditions of the unrestored salterns compared to the restored saltern or reference wetland.

It is notable that *Ms*. sp. SBSPR1A is the only genome analyzed here containing the *sdmt* and *gsmt* genes which are involved in parallel pathways for betaine biosynthesis (Figure 5 and Appendix A). The *sdmt* gene produces betaine from sarcosine, while *gsmt* produces dimethylglycine from glycine, which can then be converted to betaine with *bsmB*, a gene which was not present in any of the 11 genomes, including *Ms*. sp. SBSPR1A. Thus, it is more likely that *Ms*. sp. SBSPR1A produces betaine from sarcosine with *sdmt*, which bypasses the dimethylglycine intermediate step when synthesizing betaine from glycine [63]. While all of the genomes analyzed except *Mmv. hollandica*, and *Ml. psychrophilus* contain at least one gene for betaine transport, this additional ability to synthesize its own betaine could be an adaptation of *Ms*. sp. SBSPR1A to tolerate more hypersaline conditions than its sister genera have been found in to date. The presence of betaine synthesis and transport genes in these genomes agrees with previous work in solar salterns; on the other hand, one surprising result in these genomes is the lack of trehalose synthesis and transport genes, which were also suggested to be involved in salt tolerance in other salterns [28]. Future work is needed to identify the optimum growth conditions of this taxon. Based on the environmental sequencing data (Figure 4) and previous results from sister taxa [21], we hypothesize that while this organism can tolerate high levels of salt (e.g., ~300 ppt Cl^−^), its optimum will be at much lower concentrations.

We plan on building on the work presented here by attempting to culture the organism and run growth experiments to confirm methanogenesis substrates, as well as salt tolerance and optimum growth conditions. Such experiments will not only increase our knowledge of this organism, but also increase our understanding of the biogeochemistry and methane cycling in hypersaline ecosystems and inform the restoration of human-made systems such as industrial salt making facilities.

## 5. Conclusions

Here we have presented the draft genome sequence of *Methanosalis* sp. SBSPR1A gen. nov., sp. nov., and compared it to closely related taxa in the *Methanolobus* and *Methanomethylovorans* genera. Evidence for the discovery of a new genus includes the following: ANI values with other *Methanolobus* and *Methanomethylovorans* genomes were less than 80%, marker gene alignments place it between or adjacent to the *Methanolobus* and *Methanomethylovorans* taxa with sequenced genomes, and full length 16S rRNA gene percent identities were less than 94.5%. Similar to other *Methanolobus* and *Methanomethylovorans* species, *Ms*. sp. SBSPR1A is a methylotrophic methanogen and likely contributes to the elevated methane emissions observed in former industrial salterns [30]. *Ms*. sp. SBSPR1A has an acidic proteome similar to other halophilic organisms and is capable of synthesizing betaine, proline, ectoine, glutamate, and glutamine to tolerate environmental salinities between 97 and 232 ppt.

## Figures and Tables

**Figure 1 genes-12-01609-f001:**
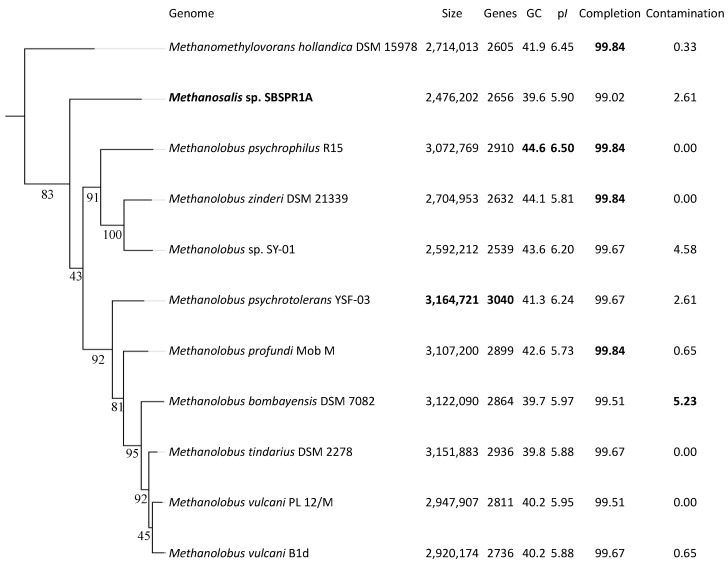
RAxML phylogenetic tree of the 11 genomes in this study using a concatenated alignment of 122 single copy archaeal genes and the PROTGAMMALG model of amino acid substitution. Branch labels show the bootstrap support, calculated with 1000 bootstraps. Also shown are the genome size, number of protein-coding genes, % G + C content, mean isoelectric point of protein-coding genes (p*I*), and percent completeness and contamination estimates from CheckM. The greatest values in each column are bolded. The tree is rooted with *Nitrosopumilus adriaticus* (Thaumarchaeota) as an outgroup (not shown).

**Figure 2 genes-12-01609-f002:**
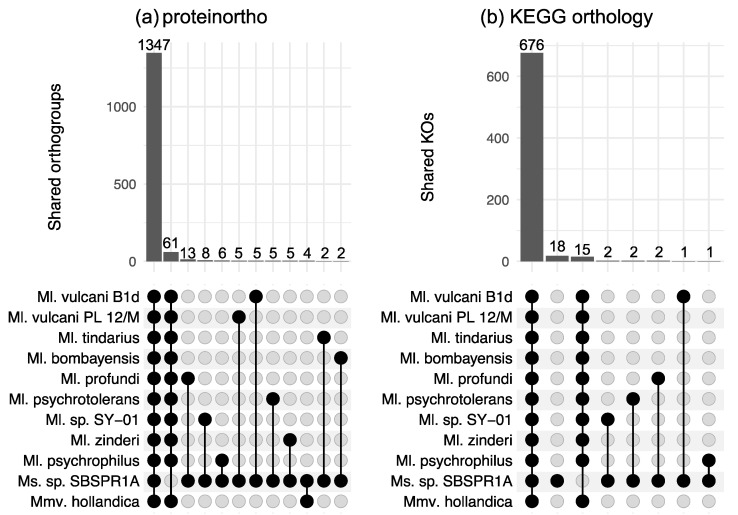
Shared orthologous gene groups among different combinations of genomes as calculated by (**a**) proteinortho, and (**b**) KEGG orthology profiles. Shown here are intersections between all 11 genomes, genes unique to *Ms*. sp. SBSPR1A (only in panel b), intersections between 10 other genomes but not *Methanosalis* sp. SBSPR1A, and pairwise comparisons between *Ms*. sp. SBSPR1A and the other genomes. Numbers above the columns state the number of shared orthogroups or shared KOs. For strain names of all of the species, see the methods section or Figure 1.

**Figure 3 genes-12-01609-f003:**
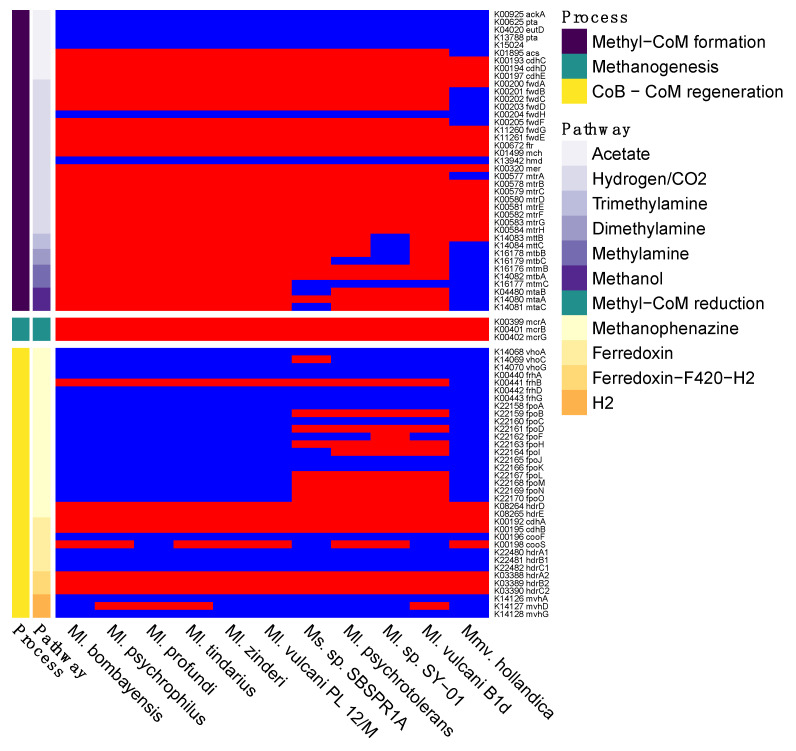
Presence (red) or absence (blue) of genes involved in methanogenesis. Shown here are six different pathways for methyl-CoM formation (Appendix A), the single methane production step of methyl-CoM reduction, and four pathways for coenzyme B (CoB) and coenzyme M (CoM) regeneration. The names of the genes and their KEGG K number assignments are shown. Columns are clustered by KO presence/absence rather than phylogeny.

**Figure 4 genes-12-01609-f004:**
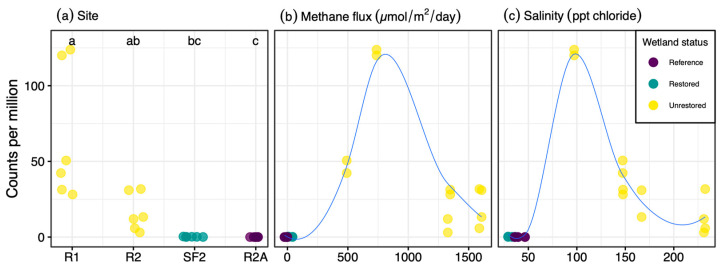
Abundance of *Methanosalis* sp. SBSPR1A, expressed as counts per million assembled reads, across 24 metagenomic samples from Zhou et al. (2021) [30], organized by (**a**) site, showing the four wetland sites sampled, (**b**) methane flux (µmol/m^2^/day), and (**c**) salinity (ppt chloride). Samples were taken from two unrestored salterns, a restored saltern, and a reference wetland. Lines in (**b**,**c**) were fitted with loess functions.

**Figure 5 genes-12-01609-f005:**
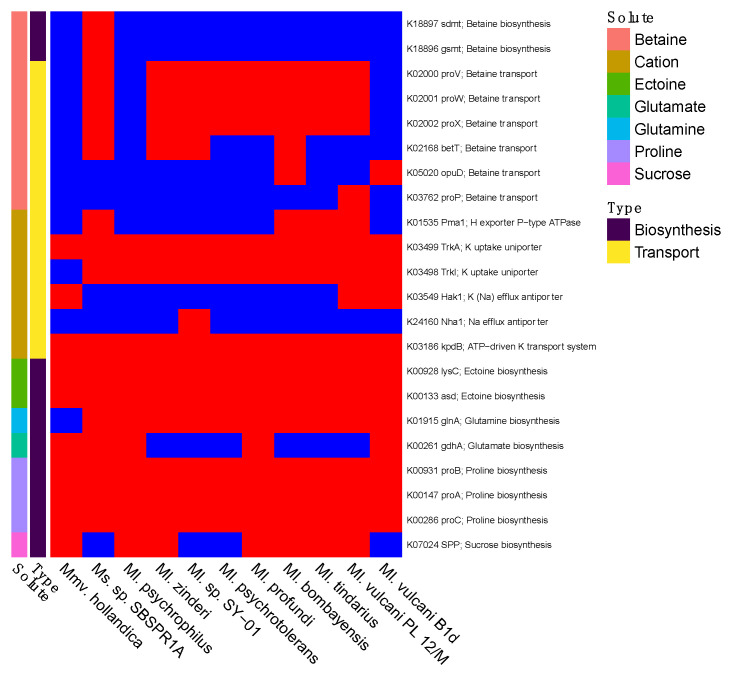
Presence (red) or absence (blue) of genes involved in compatible solute or salt biosynthesis and transport, for salt tolerance. Compounds include betaine, cations (K^+^ or Na^+^), ectoine, glutamate, glutamine, proline, and sucrose. The gene names and their KEGG K number assignments are shown. Columns were clustered according to the phylogeny in Figure 1.

## Data Availability

The new genome is publicly available with IMG ID 2929001634 and on GenBank under BioProject ID PRJNA365332 and genome accession JAIPUR000000000. The metagenome from which it was derived is also available with IMG ID 3300026157 and NCBI BioSample accession SAMN06266236. Some aspects of the analysis are publicly available on Kbase narrative ID 90991.

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
