# Peer review of "Methanogenesis and Salt Tolerance Genes of a Novel Halophilic Methanosarcinaceae Metagenome-Assembled Genome from a Former Solar Saltern"

_genes, 2021, doi:10.3390/genes12101609_

Round 1

Reviewer 1 Report

Bueno de Mesquita et al. present the first draft metagenome-assembled genome (MAG) sequence of an unclassified Methanosarcinaceae methanogen that is phylogenetically placed adjacent to the Methanolobus and Methanomethylovorans genera. The reported draft genome of 2,476,202 bp in 48 scaffolds with 2656 protein coding genes and a GC content of 39.6% is from sediments of a hypersaline unrestored industrial saltern that has been observed to be a significant methane source. The methylotrophic methanogens Methanolobus and Methanomethylovorans have not been observed in similar hypersaline environments. Phylogenetic analyses is suggestive of a new species and the genus Methanosalis is proposed. Examination of the draft genome suggested that the proposed Methanosalis SBSPR1A does not perform methanogenesis from acetate, methanol and methylamine, but only from trimethylamine and dimethylamine. The predicted proteome is acidic as would be expected from halophile and the draft genomes shows the presence of genes encoding for pathways responsible of for the synthesis and transport of compatible solutes.

Specific Comments:

(1) Fig 2: Aesthetically, the font for the species in the lower half of the figure is difficult to read. It may be just an effect of the pdf conversion. Also, the number for the intersection of all genomes on the top half of the figure would be easier to read if it were above the column.

(2) sFig 1 and Fig 3: As far as aceticlastic methanogenesis. There are two pathways that are used by methanogens in converting acetate to acetyl-CoA. Ack-Pta is utilized by the Methanosarcina sp., but acetyl-CoA synthetase (Acs, K01895) is utilized by Methanosaeta sp. Did the authors search for a gene encoding Acs? sFig1 should be modified to include Acs as a second ‘pathway for the conversion of acetate to acetyl-CoA in methanogens.

Author Response

Thank you for your prompt and positive review of our paper. Below are our responses to your comments.

(1) Fig 2: Aesthetically, the font for the species in the lower half of the figure is difficult to read. It may be just an effect of the pdf conversion. Also, the number for the intersection of all genomes on the top half of the figure would be easier to read if it were above the column.

-We updated Figure 2 accordingly. That number has been moved to above the column and all numbers enlarged. The species text has been enlarged, color changed to black, and not italicized to improve legibility.

(2) sFig 1 and Fig 3: As far as aceticlastic methanogenesis. There are two pathways that are used by methanogens in converting acetate to acetyl-CoA. Ack-Pta is utilized by the Methanosarcina sp., but acetyl-CoA synthetase (Acs, K01895) is utilized by Methanosaeta sp. Did the authors search for a gene encoding Acs? sFig1 should be modified to include Acs as a second ‘pathway for the conversion of acetate to acetyl-CoA in methanogens.

-We added the acs pathway in Figure S1 (rxn00226, rxn00176) and added K01895 as a row in Figure 3. Figure 3 now shows that acs is present in the MAG and Methanolobus genomes but not Methanomethylovorans hollandica.

Reviewer 2 Report

The work presented by Bueno de Mesquita et al. studies a genome of environmental origin belonging to a halophilic member of the Methanosarcinaceae. To this end, genomic sequencing coupled with several bioinformatic analyses, was performed. The results suggest the discovery of a new halophile whose genome contains different methane cycling genes and halotolerance genes to cope with high salt concentrations. The study overall is interesting and well presented.

Broad comments

To better follow the Materials and Methods section, please provide different subsections according to the methods described.

Specific comments

Lines 57-60: Here it should be included the citation of a work describing the identification of Methanosarcinaceae present in a salinity rhizosphere (Mirete S, et al. "Salt resistance genes revealed by functional metagenomics from brines and moderate-salinity rhizosphere within a hypersaline environment." Frontiers in Microbiology 6 (2015): 1121; doi: 10.3389/fmicb.2015.01121)

Line 95: Please, write “Methanolobus” in italics.

Line 117: Which software was used for the Kruskal-Wallis and Nemenyi posthoc tests?

Line 177: How were annotated the reactions not annotated with KOs?

Line 190, Figure S2 legend: Please, write MAG 48 Clean 2N in bold. Also, please explain the meaning of the size bar and the numbers placed at the nodes of the tree.

Line 193: Please, replace "alignment" by "phylogenetic analysis".

Line 206, Figure 1 legend: Please highlight “Ms. sp. SBSPR1A” in bold or in different color and include the genera names as they are shown in Figures S2 and S3.

Line 215: formed across? Please, check this sentence.

Lines 247-249: According to Figure 6 in the Methanosalis sp. SBSPR1A the mtmB, mtbA, and mtaA genes are present, which correspond to the methylamine and methanol pathways. Please, clarify this point.

Line 260: Does the methyl-CoM formation correspond to the hydrogenotrophic pathway mentioned in the text? Please, clarify this point.

Line 265: Please, remove "important".

Line 296, Figure 4: This figure is difficult to follow as there are several empty items. Please, check it. Also, please explain the meaning of the values present in the X-axis.

Line 313: "were clustered"

Lines 325-330: This sentence is too long. Please, split it up into two sentences.

Lines 377-392: In the context of the presence of acidic proteomes in halophiles the following article should be cited: Paul, S. et al. Molecular signature of hypersaline adaptation: insights from genome and proteome composition of halophilic prokaryotes. Genome Biol 9, R70 (2008). https://doi.org/10.1186/gb-2008-9-4-r70.

Author Response

Thank you for your prompt and positive review of our paper. Below are our responses to your comments.

Broad comments

To better follow the Materials and Methods section, please provide different subsections according to the methods described.

-We added the following subsections to the methods: 2.1. Metagenome information, 2.2. MAG assembly, 2.3. Comparative genomics

Specific comments

Lines 57-60: Here it should be included the citation of a work describing the identification of Methanosarcinaceae present in a salinity rhizosphere (Mirete S, et al. "Salt resistance genes revealed by functional metagenomics from brines and moderate-salinity rhizosphere within a hypersaline environment." Frontiers in Microbiology 6 (2015): 1121; doi: 10.3389/fmicb.2015.01121)

-Citation added, thank you for the suggestion.

Line 95: Please, write “Methanolobus” in italics.

-Fixed

Line 117: Which software was used for the Kruskal-Wallis and Nemenyi posthoc tests?

 -This was done in R with the stats and PMCMR packages. This information has been added to the text.

Line 177: How were annotated the reactions not annotated with KOs?

 -For the analysis, we only used the reactions with associated KOs. This statement refers to the fact that for some of the reactions present in the big Figure S1, the enzymes/genes are not described at all. We added the explanatory text (“…as enzymes have not yet been identified for some reactions.” to lines 185-186.

Line 190, Figure S2 legend: Please, write MAG 48 Clean 2N in bold. Also, please explain the meaning of the size bar and the numbers placed at the nodes of the tree.

 -We added a red arrow on the tree to highlight the new Methanosalis MAG. We also explained that the size bar refers to substitutions/site and the numbers refer to branch support values based on the Shimodaira-Hasegawa test.

Line 193: Please, replace "alignment" by "phylogenetic analysis".

 -Fixed

Line 206, Figure 1 legend: Please highlight “Ms. sp. SBSPR1A” in bold or in different color and include the genera names as they are shown in Figures S2 and S3.

-We updated Figure 1 to include the genus names and bolded the new Methanosalis genome.

Line 215: formed across? Please, check this sentence.

-We changed “formed across” to “identified across” to clarify this.

Lines 247-249: According to Figure 6 in the Methanosalis sp. SBSPR1A the mtmB, mtbA, and mtaA genes are present, which correspond to the methylamine and methanol pathways. Please, clarify this point.

 -That is correct, but these pathways are only partially complete. For methanol, mtaBC are missing and for methylamine mtmC is missing. We clarified this in the text in lines 259-261.

Line 260: Does the methyl-CoM formation correspond to the hydrogenotrophic pathway mentioned in the text? Please, clarify this point.

-There are 6 pathways shown here and in Figure S1 for methyl-CoM formation, including but not limited to the hydrogenotrophic pathway. We added reference to Figure S1 in this legend to help clarify.

Line 265: Please, remove "important".

-We deleted “important”

Line 296, Figure 4: This figure is difficult to follow as there are several empty items. Please, check it. Also, please explain the meaning of the values present in the X-axis.

-We are unsure what you mean by “several empty items” but we will work with the editorial office to ensure that the final figure is displayed properly. The x-axis labels are in the panel titles, but we added text to the legend to clarify this.

Line 313: "were clustered"

 -Fixed

Lines 325-330: This sentence is too long. Please, split it up into two sentences.

 -We divided this sentence into two sentences.

Lines 377-392: In the context of the presence of acidic proteomes in halophiles the following article should be cited: Paul, S. et al. Molecular signature of hypersaline adaptation: insights from genome and proteome composition of halophilic prokaryotes. Genome Biol 9, R70 (2008). https://doi.org/10.1186/gb-2008-9-4-r70.

-Citation added, thank you for the suggestion.